# The Comorbidity of Depression and Anxiety Symptoms in Tinnitus Sufferers: A Network Analysis

**DOI:** 10.3390/brainsci13040583

**Published:** 2023-03-30

**Authors:** Xuemin Chen, Lei Ren, Xinmiao Xue, Ning Yu, Peng Liu, Weidong Shen, Hanwen Zhou, Ben Wang, Jingcheng Zhou, Shiming Yang, Qingqing Jiang

**Affiliations:** 1Senior Department of Otolaryngology-Head & Neck Surgery, Chinese PLA General Hospital, Beijing 100853, China; 2National Clinical Research Center for Otolaryngologic Diseases, Beijing 100853, China; 3State Key Lab of Hearing Science, Ministry of Education, Beijing 100853, China; 4Beijing Key Lab of Hearing Impairment Prevention and Treatment, Beijing 100853, China; 5Medical School of Chinese PLA, Beijing 100853, China; 6Department of Military Medical Psychology, Air Force Medical University, Xi’an 710032, China

**Keywords:** tinnitus, depression, anxiety, network analysis, suicidal ideation

## Abstract

Objective: Sufferers of tinnitus, especially of the prolonged type, frequently suffer from comorbid depression and anxiety. From the perspective of the network model, this comorbidity is thought to be an interacting system of these two symptoms. In our study, we conducted a network analysis of depression and anxiety comorbidity in tinnitus sufferers, aiming to identify the central and bridge symptoms and make informed suggestions for clinical interventions and psychotherapy. Method: A total of 566 tinnitus sufferers were enrolled in our study. The Patient Health Questionnaire-9 (PHQ-9) and the Generalized Anxiety Disorder 7-Item Questionnaire (GAD-7) were selected to evaluate depression and anxiety symptoms, respectively, followed by network analysis to construct the interacting networks. Results: The findings identified six edges of strongest regularized partial correlations in this network. Of these, three were depression symptoms and three were anxiety symptoms. The anxiety symptoms “Unable to control worry” and “Relaxation difficulty” and the depression symptom “Feeling depressed or hopeless” had the highest expected influence centrality. The analysis results also revealed three bridge symptoms: “Afraid something awful might happen”, “Feeling of worthlessness”, and “Trouble concentrating”. As for “Suicidal ideation”, the direct relations between this symptom and “Afraid something awful might happen” and “Feeling depressed or hopeless” were the strongest. Conclusions: The central and bridge symptoms of the interacting network of depression and anxiety symptoms in tinnitus sufferers can be considered a significant transdiagnostic intervention target for the management of this comorbidity. In particular, clinical prevention and psychotherapy should be implemented, targeting the symptoms that have the strongest associations with suicidal ideation.

## 1. Introduction

Tinnitus, defined as the bothersome auditory perception in the brain or ears in the absence of external acoustical stimulus, is a worldwide complaint [1]. The prevalence of tinnitus varies widely from 5.1–42.7% around the world according to epidemiological data [2]. To date, there is no standard diagnostic criterion for tinnitus, with the most common being “tinnitus lasting over five minutes at a time” [3,4,5]. It can be the concomitant symptom of certain ear problems or systematic diseases, such as sudden sensorineural hearing loss, noise-induced hearing loss, Ménière’s disease, otitis media, presbycusis, vestibular vertigo, acoustic neuroma, diabetes, arteriosclerosis, and some psychological disorders [1,6,7]. Many tinnitus sufferers can become accustomed to the phantom sound and live in harmony with it. For others, tinnitus may lead to debilitating problems such as anxiety, depression, insomnia, frustration, irritability, concentration difficulty, and, in some extreme cases, suicide. A one year follow-up study on 386,055 tinnitus sufferers indicated that the incidence of attempted suicide was 2.06 times higher than that of propensity score-matched controls [8]. The more severe the tinnitus symptoms, the higher the prevalence of suicidal ideation [9,10,11].

It is widely believed that the injury of the cochlear ribbon synapse may cause the onset of tinnitus [12,13]. Current theories regarding the underlying mechanisms of tinnitus focus on the abnormal activities of multiple components of the peripheral and central auditory system and brain. Mechanisms such as an elevated spontaneous neuronal firing rate [14,15], the inappropriate activation of the limbic system and the central auditory cortex [16,17,18,19], increased neuronal synchronization induced by auditory deprivation [20], and changes in auditory central plasticity [21] have been implicated as potential contributors to the onset of tinnitus. High-resolution structural magnetic resonance imaging scans of subjects showed that the tinnitus sufferers exhibited reduced gray matter in the parahippocampal cortex compared to the healthy subjects [22].

Approximately 20% of tinnitus sufferers will seek clinical intervention, including pharmacological approaches [23,24], cognitive behavioral therapy [23,25], sound therapy [26,27], biofeedback therapy [28], and masking devices [29]. There is a positive correlation between the tinnitus severity index and neuroticism scores, as well as anxiety and depressive scores, which insinuates that the factors closely related between tinnitus and depression and anxiety may fundamentally lie in personality traits [30,31]. Previous population-based evidence has shed light on the relationships between tinnitus symptoms and mental disorders, and psychiatric comorbid depression and anxiety have been reported as a potential modulators of brain structural changes in tinnitus sufferers [32]. Bhatt et al. [33] carried out a tentative exploration of the relationships between tinnitus and the prevalence of anxiety and depression. They found that 25.6% and 26.1% of tinnitus sufferers reported problems of depression and anxiety, respectively, while only 9.1% and 9.2% of those without tinnitus reported corresponding symptoms. Similarly, Belli et al. [34] concluded that chronic tinnitus sufferers had significantly higher Beck Depression Inventory and Beck Anxiety Inventory scores. Notably, tinnitus severity has been correlated with depression and anxiety levels, and with improvements in depression and anxiety symptoms, the tinnitus prevalence may decrease [35,36]. It is also conceivable that depression and anxiety symptoms may precede tinnitus onset and predispose for it [37]. Depression and anxiety are bidirectional risk factors for one another, and the presence of one symptom of depression or anxiety often triggers the onset of the other [38,39,40]. These comorbidities and sequelae should be recognized and addressed to optimally treat sufferers with bothersome tinnitus. Hence, it is well worth conducting further study into the relationships among tinnitus, depression, and anxiety in order to improve the wellbeing and quality of life of tinnitus sufferers.

There are many psychological models to study the correlation between depression and anxiety and organic diseases. In recent years, the data-driven network model, a framework to conceptualize psychological disorders, has been utilized. This model can analyze and intuitively display the relationships between multiple variables [41,42]. The network usually consists of nodes and edges. Nodes refer to the variable, or rather, the symptoms of mental disorders, while edges refer to the connections between the variables [43]. From a network point of view, psychiatric symptoms are not mental diseases in essence, but components of mental diseases. Mental disorders including anxiety and depression arise from the direct interactions between psychiatric symptoms, rather than a single underlying latent variable [44]. Network analysis can build a structural network, reflecting not only the weight of each item of the clinical questionnaire, but also the complex interrelationship between different items. The network is helpful to identify which central symptoms determine the severity of the mental disorder and deserve the utmost clinical attention and timely intervention. In addition, the concept of bridge symptoms, which may increase the risk of other disorders [45], also provides new insights into the emergence of comorbidities and inspirations for intervention.

In the current study, for the first time, we applied a verified network model to examine the network structure of depression and anxiety symptoms in tinnitus sufferers, and to identify the central and bridge symptoms as targets for clinical or psychological interventions. Of note, we focused on the symptoms directly related to “Suicidal ideation”, with the aim of reducing the suicide rate through early intervention.

## 2. Methods

### 2.1. Participants

A total of 566 tinnitus sufferers aged over 11 years old, followed up at the tinnitus outpatient department in Chinese PLA General Hospital (Beijing, China), were recruited in this study from December 2021 to June 2022. The exclusion criteria included: (1) declined or ignored the recruitment; (2) did not complete the questionnaire or missed items; (3) no tinnitus. This study was approved by the Ethics Committee of Chinese PLA General Hospital (No. S2021-179-02). All participants or their legal guardians provided their written informed consent to participate.

### 2.2. Evaluating Depression and Anxiety Symptoms

The Patient Health Questionnaire (PHQ-9) and Generalized Anxiety Disorder-7 (GAD-7) questionnaires used in this study are the quantitative evaluation criteria recommended by the Diagnostic Statistical Manual of Mental Disorders (DSM-V) published by the American Psychiatric Association. These two questionnaires have simple content and strong maneuverability, and have been proven to have good reliability and validity [46,47].

The PHQ-9 is a simple and efficient self-assessment tool for evaluating depressive symptoms over the past two weeks [48]. It consists of nine items, and the answers to each item are composed of four options, namely, not at all (0 points), several days (1 point), more than half the days (2 points), and almost every day (3 points). The maximum total score is 27, which is positively correlated with the degree of depression. A score of 0 to 4 is considered normal, 5 to 9 as mild depression, 10 to 14 as moderate depression, 15 to 19 as moderately severe depression, and 20 to 27 as severe to profound depression.

GAD-7 is a seven-item reliable self-assessment questionnaire designed for screening the aspects and frequency of anxiety symptoms over the previous two weeks [49]. The options and scores are the same as those of the PHQ-9. The maximum total score is 21. Based on the final total score, anxiety can be classified as normal (0–4), mild (5–9), moderate (10–14), and severe (15–21). 

### 2.3. Network Analysis

#### 2.3.1. Network Estimation and Visualization

Gaussian graphical models (GGM) were used to estimate the networks via the R package *qgraph* [50]. It is a non-oriented network whose edge represents the partial correlation between two nodes after all of the other nodes have been controlled for [51]. The GGM was estimated in the light of nonparametric Spearman’s rho correlation matrices (Appendix A). The graphical least absolute shrinkage and selection operator (LASSO) algorithm was used for regularization [52], which shrinks all edges and sets those with small partial correlations to zero, so that a simplified non-oriented weighted network is obtained. The tuning parameter was set to 0.5, so that the sensitivity and specificity for determining the true edges could be well balanced [53]. The visualization of the network was derived from the Fruchterman–Reingold algorithm [54], which locates the nodes with stronger correlations near the center of the network and weaker correlations on the periphery. In these networks, blue edges represent positive partial correlations, while red edges represent negative partial correlations. The thicker the edges, the stronger the partial correlations between the nodes.

#### 2.3.2. Expected Influence and Predictability Analysis

The node expected influence was calculated via R-package *qgraph* [55]. Expected influence refers to the sum of the values of all edges connecting to a particular node. Higher expected influence values mean more significance in the network. Moreover, the node bridge expected influence was calculated via R-package *networktools* [56]. Bridge expected influence refers to the sum of the values of all edges connecting a particular node to nodes in the other community. The higher the bridge expected influence, the higher the possibility of contagion to other communities. In this study, the nodes were classified into two communities according to the questionnaire used (PHQ-9 and GAD-7). Bridge symptoms were identified using a blind 80th percentile cut-off on the bridge strength scores [56]. In addition, the predictability of the nodes was calculated via R-package *mgm* [57]. Predictability refers to the degree to which the variance of a node can be explained by all of its connected nodes, which can characterize the controllability of the network. High node predictability suggests that we can control it through its neighbors, whereas low node predictability implies that we can intervene directly with the node itself or look for other variables outside the network to control it. 

#### 2.3.3. Network Accuracy and Stability

The stability of the network was evaluated by conducting the R-package *bootnet* algorithm [58]. Firstly, we evaluated the stability of the edges in the network by resampling and replacing the observed values (nboots = 2000), so as to obtain a 95% confidence interval (CI) for each sample value. Edges with less overlap with the 95% CI of other edges were suggested to be irreplaceable, which is more important in maintaining the stability of the network. Secondly, the case-dropping bootstrap method (nboots = 2000) was used to calculate the correlation stability coefficient (CS) in order to evaluate the stability of the expected influences and bridge expected influences. CS refers to the maximum proportion of nodes that can be deleted under the condition that the correlation with the original centrality index reaches or exceeds 0.7. A CS greater than 0.5 indicates good node stability, and the minimum CS should not be less than 0.25 [58]. Thirdly, bootstrapped difference tests (nboots = 2000, α = 0.05) were used to detect whether there were significant differences in the edge weights, nodes expected influences and bridge expected influences. 

## 3. Results

### 3.1. Basic Descriptive Characteristics of Depression and Anxiety Severity in Tinnitus Sufferers

There were 566 tinnitus sufferers enrolled. Among them, 284 were male (50.18%) and 282 were female (49.82%). The mean age was 44.74 ± 14.33 years (means ± SD, range 11–86 years). The PHQ-9 and GAD-7 scores reflect the severity of the comorbidity symptoms of the participants. Detailed demographic characteristics and the number and percentage of depression and anxiety symptom severity in the enrolled tinnitus sufferers are listed in Table 1. Table 2 shows the mean scores, SDs, and predictability for each item of the PHQ-9 and GAD-7 questionnaire. Appendix A shows the nonparametric Spearman’s rho correlation matrix of these symptoms.

### 3.2. Network Structure of Depression and Anxiety Symptoms in Tinnitus Sufferers

The internal consistency of PHQ-9 and GAD-7 in this study was acceptable (α = 0.90 and 0.91, respectively). The network of depression and anxiety comorbidity in tinnitus sufferers is illustrated in Figure 1a. Of the 120 possible edges in this network, 82 edges were not zero (68%). All of these edges were positive except the edge between D3 “Sleep problems” and D6 “Feeling of worthlessness” (weight = −0.04), and between D4 “Energy loss” and D9 “Suicidal ideation” (weight = −0.02). Then, we found the six strongest edges in the network, that is, D1 “Anhedonia” and D2 “Feeling depressed or hopeless” (weight = 0.33), D1 “Anhedonia” and D4 “Energy loss” (weight = 0.27), D3 “Sleep problems” and D5 “Appetite changes” (weight = 0.22), A2 “Unable to control worry” and A3 “Excessive worry” (weight = 0.24), A2 “Unable to control worry” and A5 “Too restless to sit still” (weight = 0.22), and A4 “Relaxation difficulty” and A5 “Too restless to sit still” (weight = 0.22). Notably, none of these strongest edges linked anxiety and depressive symptoms.

The bootstrapped 95% CI indicates that the accuracy of the edge weights is relatively reliable and accurate (Appendix A). In addition, the bootstrapped difference test for the edge weights indicated that the six strongest edge weights differed significantly from the other edge weights, at ratios of approximately 44% to 85% (Appendix A). In addition, the node predictability was visualized, as shown in Figure 1a. The value of the node predictability ranged from 32% to 70%; among these, anxiety symptom A2 “Unable to control worry” had the highest predictability, while depression symptom D3 “Sleep problems” had the lowest predictability (Table 2). The average was 53%, indicating that, on average, 53% of the node variances can be explained by their neighboring nodes. 

The expected influences of the current network are shown in Figure 1b. The anxiety symptoms A2 “Unable to control worry”, A4 “Relaxation difficulty”, and depression symptom D2 “Feeling depressed or hopeless” had the highest expected influences, indicating that these three symptoms are the most relevant symptoms in this network. Likewise, depression symptom D9 “Suicidal ideation” had the lowest expected influences, indicating that this symptom is the least associated symptom. The CS of the node expected influence was 0.60, illustrating that our estimates were adequately stable (Appendix A). Further, the bootstrapped difference tests for node expected influences showed that the expected influences of the three symptoms with highest expected influences were significantly different to the other symptoms, at ratios of approximately 33% to 73% (Appendix A).

### 3.3. Bridge Symptoms of Depression and Anxiety in Tinnitus Sufferers

Figure 2b presents the results of the bridge expected influence. Based on the data, we identified anxiety symptom A7 “Afraid something awful might happen” and depression symptoms D6 “Feeling of worthlessness” and D7 “Trouble concentrating” as bridge symptoms. These selected bridge symptoms are labeled with yellow color in Figure 2a. This indicated that anxiety symptom A7 “Afraid something awful might happen” had the strongest possibility of contagion to the depression community. Conversely, depression symptoms D6 “Feeling of worthlessness” and D7 “Trouble concentrating” had the strongest ability to increase the risk of contagion to the anxiety community. The CS of the node bridge expected influence was equal to 0.36, indicating that our estimates of node bridge expected influences met the requirement (Appendix A). Moreover, the bootstrapped difference tests for the node bridge expected influences showed that the bridge expected influences of the three bridge symptoms were significantly different to the other symptoms at an average ratio of 27% (Appendix A).

### 3.4. Flow Network of Suicidal Ideation with Depression and Anxiety Symptoms in Tinnitus Sufferers

Figure 3 delineated a flow chart showing how depression symptom D9 “Suicidal ideation” was connected to all other symptoms in the current network. There were nine symptoms directly related to D9 “Suicidal ideation” and six symptoms indirectly related to it. The direct relations between D9 “Suicidal ideation” and anxiety symptom A7 “Afraid something awful might happen” and depression symptom D2 “Feeling depressed or hopeless” were the strongest. Additionally, the predictability of depression symptom D9 “Suicidal ideation” indicated that 41% of its variance could be explained by its neighboring nodes (Table 2).

## 4. Discussion

In the present study, we explored the network structure of depression and anxiety comorbidity in 566 tinnitus sufferers using network analysis. We found that the strongest edges exist within each disorder, which is in line with other network research focusing on the comorbidity of depression and anxiety symptoms [59,60,61,62,63,64]. Our findings indicated that six edges with the strongest regularized partial correlations existed in the network. Three were among depression symptoms and three were among anxiety symptoms.

Among all of the symptoms, the anxiety symptom A2 “Unable to control worry” had the highest expected influence centrality, followed by the anxiety symptom A4 “Relaxation difficulty” and the depression symptom D2 “Feeling depressed or hopeless”. The high expected influence centrality indicated that alleviating these nodes may destabilize the psychopathological networks and confer the highest general benefit to ease mental distress [42]. Some interventions could be implemented to alleviate these three symptoms. For example, A2 “Unable to control worry” was identified as the core symptom of the tinnitus sufferers’ network in this study, which is consistent with the findings of Natalini and colleagues’ study [65]. The authors developed the innovative idea that psychotherapeutic approaches could focus on alterations to the metacognitions of tinnitus sufferers. The concept of metacognition refers to “the aspect of information processing that monitors, interprets, evaluates, and regulates the contents and processes of its organization” [66]. Metacognitive therapy (MCT) is based on the principle that the vulnerability of those with mental disorders is connected to cognitive attentional syndrome (CAS). CAS is a pattern of repetitive negative thinking in the process of worrying and ruminating. MCT focuses on raising awareness of rumination and removing the behavior, developing new responses to negative thoughts, and improving selective attention control [67,68]. Ferraro et al. evidenced the therapeutic impact of MCT on the perception of tinnitus and its anxiety and depression correlates [69]. Regarding A4 “Relaxation difficulty” in anxiety disorders, mindfulness meditation and relaxation therapies have been shown to be effective in diminishing the effects of tinnitus and can eventually guide sufferers towards habituation [70,71]. The specific operational methods are referred to in a previous article [72]. Mindfulness meditation is a technique designed to help one purposely pay attention to the present experience non-judgmentally [73]. Relaxation therapies include various methods such as progressive relaxation, differential relaxation, cue-controlled relaxation, rapid relaxation, and biofeedback therapy [74], all of which are easy to learn. As for the depressive symptom D2 “Feeling depressed or hopeless”, cognitive behavioral therapy (CBT) might be a good choice for tinnitus sufferers. CBT is a psychological intervention that aims to alleviate depression or low mood by helping sufferers to modify their dysfunctional cognition, ruminations, and safety-seeking behaviors [75]. A large and credible randomized controlled trial [76] and meta-analysis [77] have indicated the beneficial effects of CBT on tinnitus-related distress.

Our results also revealed three bridge symptoms: A7 “Afraid something awful might happen”, D6 “Feeling of worthlessness”, and D7 “Trouble concentrating”. As A7 “Afraid something awful might happen” had the highest bridge strength centrality among all of these symptoms, it might be considered a significant transdiagnostic intervention target for the prevention of both mental disorders in tinnitus sufferers. It was also the highest bridge strength among all of the anxiety symptoms in the GAD-7, which indicates that it has the greatest liability for increasing the risk of contagion to depression, and thus deserves the utmost attention. Conversely, when depression is present, treating D6 “Feeling of worthlessness” and D7 “Trouble concentrating” may decrease the risk of contagion to anxiety. Treatments include the abovementioned CBT or relaxation therapies. In addition, transcranial direct current stimulation (tDCS) and repetitive transcranial magnetic stimulation (rTMS) have also been proven promising as potent therapeutic tools for tinnitus, which can be used as neuroregulatory interventions to alleviate the symptoms of mental disorders [78,79].

Previous studies have revealed that tinnitus sufferers are at increased risk of suicide [80,81,82]. Suicidal ideation and initial attempts often precede actual suicide completion. According to a national survey with 17,466 participants in Korea [83], a total of 20.9% and 1.2% of tinnitus sufferers, and 12.2% and 0.6% of those without, reported suicidal ideation and attempts, respectively (*p* < 0.0001 and *p* = 0.001). Therefore, in order to decrease the rates of completed suicide, it is of considerable significance to identify the warning signs and risk factors associated with suicidal ideation and attempts. In this network, we found that the direct relation between D9 “Suicidal ideation” and A7 “Afraid something awful might happen” was the strongest, followed by D2 “Feeling depressed or hopeless”. Most of the previous studies focused on the correlation between depression and suicide [84,85], for the reason that depression tends to lead to loneliness, lack of interest, self-deprecation, self-hatred, etc., while the correlation between anxiety and suicide is rarely mentioned. The association between prospective suicide attempts and comorbid anxiety disorders have recently been evidenced by several studies [60,86,87,88]. Uniquely, A7 “Afraid something awful might happen” has not previously been highlighted, to our knowledge. There is evidence that olanzapine co-administered with mood stabilizers in patients with agitated mixed states can decrease suicidal ideation [89,90]. Thus, alleviating the feeling of endless fear with psychotherapy or anxiolytics and antidepressants might be conducive as an intervention regarding suicide in tinnitus sufferers. 

There are some limitations to our study. First, the network structure is specific to the questionnaires we used in this study, which means that different assessment questionnaires might lead to different network structures. Second, given that the data we obtained were cross-sectional, we cannot identify the directionality of the edges, or rather, the causality between the most central node and other nodes. Longitudinal data are needed in future studies. Third, the results of network analysis are affected by factors such as the number of included nodes and the calculation methods, and the results need to be interpreted reasonably in combination with clinical practice. Fourth, further investigations are required to evaluate the potential influence of depressive and anxiety symptoms on this estimated network. Fifth, some items of the questionnaire are ambiguous. For example, D5 “Appetite changes” can mean either eating more or less, while D8 “Psychomotor issues” can mean either agitation or retardation. Finally, a larger sample of psychological data is needed. 

## 5. Conclusions

The current study highlighted three critical central symptoms (“Unable to control worry”, “Relaxation difficulty”, and “Feeling depressed or hopeless”) and three bridge symptoms (“Afraid something awful might happen”, “Feeling of worthlessness”, and “Trouble concentrating”) within the interacting network of depression and anxiety in tinnitus sufferers, which might be considered a significant transdiagnostic intervention target for the prevention of comorbidity. In particular, clinical prevention and psychotherapy should be implemented that targets the symptoms with the strongest associations with suicidal ideation (“Afraid something awful might happen” and “Feeling depressed or hopeless”).

## Figures and Tables

**Figure 1 brainsci-13-00583-f001:**
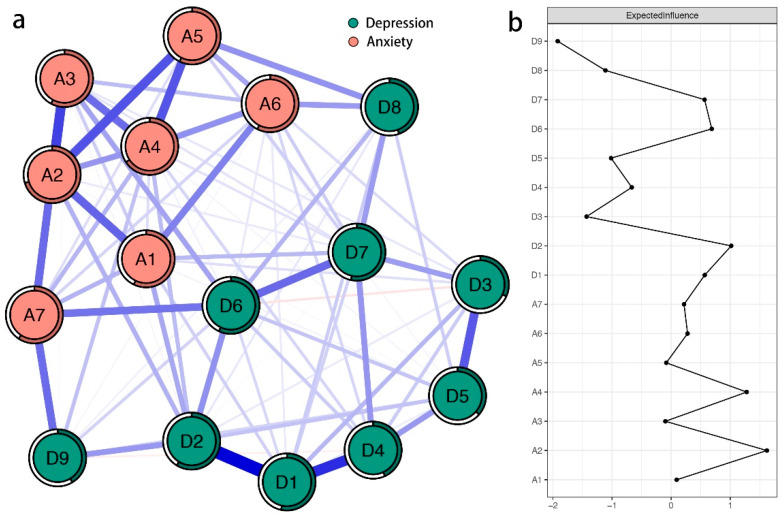
Network structure of depression and anxiety symptoms in tinnitus sufferers. (**a**) Blue edges represent positive correlations, red edges represent negative correlations, and the thickness of the edge reflects the magnitude of the correlation. The circles around nodes depict their predictability. (**b**) Centrality plot depicting the expected influence of each symptom in the network (z-score). D1 = Anhedonia; D2 = Feeling depressed or hopeless; D3 = Sleep problems; D4 = Energy loss; D5 = Appetite changes; D6 = Feeling of worthlessness; D7 = Trouble concentrating; D8 = Psychomotor issues; D9 = Suicidal ideation; A1 = Feeling nervous or anxious; A2 = Unable to control worry; A3 = Excessive worry; A4 = Relaxation difficulty; A5 = Too restless to sit still; A6 = Easily annoyed or irritable; A7 = Afraid something awful might happen.

**Figure 2 brainsci-13-00583-f002:**
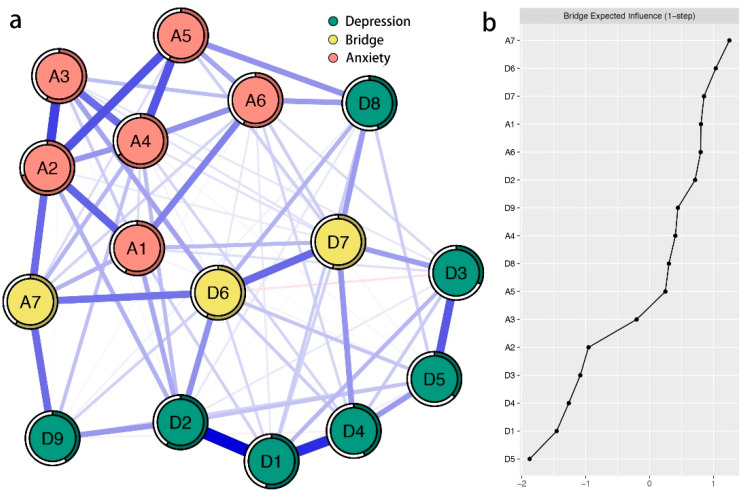
Network structure of depression and anxiety symptoms showing bridge symptoms in tinnitus sufferers. (**a**) Blue edges represent positive correlations, red edges represent negative correlations, and the thickness of the edge reflects the magnitude of the correlation. The circles around nodes depict their predictability. (**b**) Centrality plot depicting the bridge expected influence of each symptom in the network (z-score). D1 = Anhedonia; D2 = Feeling depressed or hopeless; D3 = Sleep problems; D4 = Energy loss; D5 = Appetite changes; D6 = Feeling of worthlessness; D7 = Trouble concentrating; D8 = Psychomotor issues; D9 = Suicidal ideation; A1 = Feeling nervous or anxious; A2 = Unable to control worry; A3 = Excessive worry; A4 = Relaxation difficulty; A5 = Too restless to sit still; A6 = Easily annoyed or irritable; A7 = Afraid something awful might happen.

**Figure 3 brainsci-13-00583-f003:**
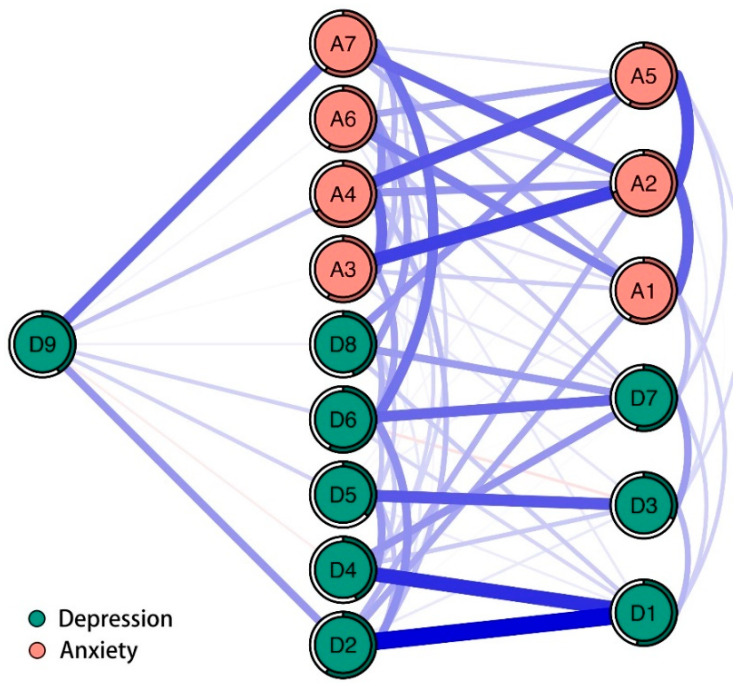
Flow network of suicidal thoughts. Blue edges represent positive correlations, red edges represent negative correlations. The thickness of the edge reflects the magnitude of the correlation. The circles around nodes depict their predictability. D1 = Anhedonia; D2 = Feeling depressed or hopeless; D3 = Sleep problems; D4 = Energy loss; D5 = Appetite changes; D6 = Feeling of worthlessness; D7 = Trouble concentrating; D8 = Psychomotor issues; D9 = Suicidal ideation; A1 = Feeling nervous or anxious; A2 = Unable to control worry; A3 = Excessive worry; A4 = Relaxation difficulty; A5 = Too restless to sit still; A6 = Easily annoyed or irritable; A7 = Afraid something awful might happen.

**Table 1 brainsci-13-00583-t001:** Demographic characteristics and the number and percentage of depression and anxiety symptom severity of enrolled tinnitus sufferers.

Variables	Mean	SD
**Age**	44.74	14.33
	**Number**	**Percentage**
**Gender**		
Male	284	50.18%
Female	282	49.82%
**Education**		
Junior high school and below	107	18.90%
Senior high school or technical secondary school	107	18.90%
Junior college	114	20.14%
Undergraduate	179	31.63%
Master	51	9.01%
Doctor	8	1.41%
**Duration of tinnitus**		
<3 months	202	35.69%
3–6 months	63	11.13%
6–12 months	71	12.54%
>12 months	230	40.64%
**Family history**		
Yes	90	15.90%
No	476	84.10%
**Tinnitus perception at onset**		
Gradual	177	31.27%
Abrupt	389	68.73%
**Tinnitus location**		
Right ear	166	29.33%
Left ear	177	31.27%
Both ears	209	36.93%
Inside the head	14	2.47%
**Tinnitus manifestation**		
Intermittent	121	21.38%
Constant	445	78.62%
**Visual analog scale (VAS) score**		
Minimal symptoms (range = 0–3)	186	32.86%
Mild symptoms (range = 4–6)	217	38.34%
Severe symptoms (range = 7–10)	163	28.80%
**Depression symptoms (PHQ-9)**		
Minimal symptoms (range = 0–4)	241	42.58%
Mild symptoms (range = 5–9)	164	28.96%
Moderate symptoms (range = 10–14)	102	18.02%
Moderately severe symptoms (range = 15–19)	39	6.89%
Severe symptoms (range = 20–27)	20	3.53%
**Anxiety symptoms (GAD-7)**		
Minimal symptoms (range = 0–4)	288	50.88%
Mild symptoms (range = 5–9)	169	29.86%
Moderate symptoms (range = 10–14)	68	12.01%
Severe symptoms (range = 15–21)	41	7.24%

SD = Standard Deviation; PHQ-9 = Patient Health Questionnaire-9; GAD-7 = Generalized Anxiety Disorder-7.

**Table 2 brainsci-13-00583-t002:** Mean scores, standard deviations, and predictability for each symptom of the PHQ-9 and GAD-7.

Symptoms	Mean	SD	Predictability
**Depression symptoms (PHQ-9)**	6.83	5.65	
PHQ-1: Anhedonia (D1)	0.99	1.01	0.53
PHQ-2: Feeling depressed or hopeless (D2)	0.84	0.94	0.59
PHQ-3: Sleep problems (D3)	1.2	1.05	0.32
PHQ-4: Energy loss (D4)	1.15	1.03	0.43
PHQ-5: Appetite changes (D5)	0.58	0.83	0.36
PHQ-6: Feeling of worthlessness (D6)	0.55	0.82	0.57
PHQ-7: Trouble concentrating (D7)	0.78	0.92	0.53
PHQ-8: Psychomotor issues (D8)	0.56	0.82	0.45
PHQ-9: Suicidal ideation (D9)	0.18	0.47	0.41
**Anxiety symptoms (GAD-7)**	5.52	5.12	
GAD-1: Feeling nervous or anxious (A1)	0.94	0.91	0.57
GAD-2: Unable to control worry (A2)	0.77	0.90	0.70
GAD-3: Excessive worry (A3)	0.95	0.95	0.58
GAD-4: Relaxation difficulty (A4)	0.89	0.94	0.66
GAD-5: Too restless to sit still (A5)	0.52	0.80	0.57
GAD-6: Easily annoyed or irritable (A6)	0.97	0.94	0.57
GAD-7: Afraid something awful might happen (A7)	0.48	0.76	0.59

SD = Standard Deviation; PHQ-9 = Patient Health Questionnaire-9; GAD-7 = Generalized Anxiety Disorder-7.

## Data Availability

The raw data supporting the conclusions of this article will be made available from the corresponding author on reasonable request.

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
