# Peer review of "The Comorbidity of Depression and Anxiety Symptoms in Tinnitus Sufferers: A Network Analysis"

_brainsci, 2023, doi:10.3390/brainsci13040583_

Round 1

Reviewer 1 Report

In their article "The comorbidity of depression and anxiety symptoms in tinnitus sufferers: a network analysis" , the authors address two of the most common comorbidities of tinnitus sufferers - anxiety and depression.

For this purpose, they use the statistical tool of network analysis to find out possible influencing variables. In addition to the basic analysis of possible connections between the single variables, the focus was on the variable "suicidality" in order to be able to intervene therapeutically at an early stage.

Strengths:

One strength of the article is the statistical approach that analyzes all individual questions of the two screening instruments that were used. This can highlight possible cross-links and key psychiatric/psychological relevant symptoms.In the discussion, clear indications of therapeutic options are given. In this respect, a basic statistical tool has a direct clinical impact.

Shortcomings

It is not understandable why the authors put so much increased emphasis on the red-flag symptom "suicidal tendency"

From the list of symptoms available for selection, this is certainly the most serious symptom, which is also of particular importance in everyday clinical practice. Nevertheless, essential epidemiological data on the frequency of suicidality are missing. The network data collected also do not justify the special role of this item.
The article could easily dispense with this focus and instead interpret the main nodes or edges found in an open and unbiased manner.

If suicidality is considered essential, the authors should cite more extensive epidemiologic data in the introduction.
Furthermore, a more comprehensive presentation of the concepts "anxiety" and "depression" from a network theoretical point of view would be desirable in the introduction - there are some references to this in the discussion, but there is certainly more background knowledge (independent of tinnitus!).

Minor:

The introduction to the discussion (lines 269-283) is superfluous or could supplement the information in the introduction

Author Response

Thanks for your suggestion. Firstly, we have added relevant statistical data to support the conclusion that suicidal ideation may be more prevalent in tinnitus sufferers (Line 52-55). In this study, we put our emphasis on the suicidal ideation, because the incidence of attempted suicide in the tinnitus sufferers group was 2.06 times higher than that of propensity score-matched controls. Besides, we rearranged the introduction part according to your suggestion, and delete the superfluous part in the discussion.

Reviewer 2 Report

Thank you for the opportunity to review this paper. This paper falls well within my expertise, I have a background in applied statistics and behavioral assessment in clinical, non-clinical and in translational animal model setting. In this study, the authors resent a network modeling examination of anxiety and depression in tinnitus sufferers. The study focuses I suicidal ideation with the motivation of finding targets for the development of interventions.

The paper is very well written, the language is clear and the narrative flows well and follows a logical sequence. The study uses very popular and well validated anxiety and depression tools. I like how the network analysis is presented with the robustness assessments done with the bootstrap. I believe overall this work is very well done. I agree with the limitations although the concerns expressed on the ambiguity of some of the questions may be too harsh since changes in D5 and D8 in either direction do not necessarily change the interpretation.

I have two small concern/comments that I believe should be addressed:

First, it would be useful to add more context by providing more on the demographic and clinical descriptors of the sample used. The only descriptors are in lines 173-177. In comparison Natalini et al 2020 (cited here as reference 61), do provide excellent descriptive tables.

Second, on the bridge symptoms, although there is some discussion about it in lines 323-335, I wonder if the evidence here provides any peculiarities that are relevant to tinnitus sufferers. In other words, what are the bridge symptoms in non-tinnitus sufferers? and, what is unique to tinnitus sufferers?

Based on my assessment I would recommend a minor revision for this article, as I believe my concerns/comments can be properly addressed in a simple revision.

Congratulations on the good work and good luck on your careers.

Author Response

Thanks for your suggestion. Firstly, we have revised the Table 1 according to your suggestions, and added detailed demographic characteristics of tinnitus sufferers enrolled. Then, for the first time, we applied verified network model to examine the network structure of depression and anxiety symptoms in tinnitus sufferers. Although we did not construct a network in non-tinnitus sufferers, we found that the network in tinnitus sufferers was relatively stable and unique. These three bridge symptoms were calculated and verified based on tinnitus sufferers’ data from our outpatient, and databases with larger sample sizes are needed for further validation and generalization.